# Determining acute nurse staffing: a hermeneutic review of an evolving science

Alison Leary,[1,2] Geoffrey Punshon[1]

[1]School of Health and Social Care, London South Bank University, London, UK
[2]School of Health, University of South Eastern Norway, Oslo, Norway

**Correspondence to**
Professor Alison Leary;
alisonleary@yahoo.com

## ABSTRACT

**Background** Calculating nurse staffing in the acute hospital has become a key issue but solutions appear distant. Community, mental health and areas such as learning disability nursing have attracted less attention and remain intractable. This review aims to examine current approaches to the issue across many disciplines.

**Design** The approach taken is iterative and in the form of a hermeneutic review. 769 pieces of evidence were reviewed from across disciplines such as nursing, medicine, engineering, statistics, population science, computer science and mathematics where hospital nurse staffing was the subject of the study.

**Results** A number of themes emerged. The first iteration showed the predominance of unit base approaches (eg, nurse numbers, ratios, activity and workload) and the second was the development of methodologies. Subsequent iterations examined issues such as demand, safety, nurse education, turnover, patient outcomes, patient or staff satisfaction, workload and activity. The majority of studies examined (n=767) demonstrated some association between staffing (units or type/skill) and various factors such as staff or patient satisfaction, working conditions, safety parameters, outcomes complexity of work achieved, work left undone or other factors. Many potential areas such as operational safety research were not utilised.

**Conclusion** Although the relationship between staffing in acute care and factors such as units, safety or workload is complex, the evidence suggests an interdependent relationship which should only be dismissed with caution. The nature of these relationships should be further examined in order to determine nurse staffing. The body of knowledge appears substantial and complex yet appears to have little impact on policy.

## Strengths and limitations of this study

► This is a review of the existing literature across many fields and focuses on cross-disciplinary approaches.
► By encompassing many fields breadth rather than depth limits the analysis.
► This study suggests looking at the issue of safe staffing from a wider operational safety perspective could uncover additional insight and solutions.

## INTRODUCTION

The question of registered nurses (RNs) staffing hospitals to a level, that is, safe, high quality and cost effective has become a key issue worldwide. In terms of research, it has also become something of a Gordian knot— one that attracts many attempts to solve but few solutions. Extending this question to community nursing and areas such as mental health or learning disability nursing demonstrate the complexity of the problem. Fundamental questions such as 'How many nurses are needed?' 'What kind of skill mix?' 'Are staffing ratios the solution?' remain largely unanswered.

These issues present interesting theoretical problems but they have a very real-world application as they underpin the safety of patients both in hospital and community. It could be argued that nursing is only visible by its absence such as those reported by Francis in the enquiry into the deaths at Mid Staffordshire NHS Foundation trust.[1] Nursing is the largest part of the healthcare workforce[2] and attempts to curtail costs have seen radical workforce changes including reducing the number of RNs or replacing them with assistive personnel.[3] It is of interest that there have been a number of coroners commenting on staffing levels under section 28 of the Coroners & Justice Act 2009[4]. This allows a Coroner in England to report circumstances where it may prevent further deaths. There are several examples of this for both the acute and care home sectors in recent years.[5]

Issues of staffing are further complicated by a lack of consensus within the profession around the levels of care that should be provided or calculated. Is there a difference between safe care and good care? Should all care be given by a RN? Certainly, policy decisions in England are promoting the transfer of care to assistive personnel such as nursing associates[6] for a variety of reasons such as shortage of RNs and increasing costs. Financial burden is a primary driver of workforce determinants. Investing in one aspect of

staffing often means scaling back elsewhere, resulting in 'trade-offs', for example, using less educated staff or changing nurse patient ratios.[7]

Different countries within the UK are taking different policy positions on staffing which varies from guidance in England to legislation in Scotland and Wales.[8]

In order to make sense of the variation in published work, a hermeneutic approach is taken. Such an approach is to question and to remain open to what might be revealed.[9] Different researchers have used different approaches to understand various aspects of this problem and the aim of this work to reveal a deeper understanding and understand more about the interrelational nature of the problem. This paper reviews the approaches that have been used to investigate the idea of safe staffing, the knowledge elicited and explores direction this area of research might take in the future.

## METHOD

Hermeneutic reviews[10] use a process of searching and interpretation as interrelational activity (figure 1). This is because the focus is on the understanding and interpretation of the materials. In a subject with a high volume of literature, this approach allows the integration of interpretation and analysis of the literature and the development of searching as part of the review. A

hermeneutic framework describes the literature review process as fundamentally a process of developing understanding, that is, iterative in nature.[10] This has allowed a much wider perspective incorporating a wider search and analysis than, for example, a systematic review utilising methods such as Preferred Reporting Items for Systematic Reviews and Meta-Analyses.[11] This body of evidence uses an extremely diverse set of methodologies ranging from inductive studies reliant on qualitative data to areas of computational mathematics. This would make approaches such as meta-analysis almost impossible.

Initial reading indicated that the work on this topic was not limited to the nursing literature and so the first iteration was to expand the search to areas such as computer science, maths and engineering. A search of the common databases was undertaken including those in informatics and computational mathematics (PubMed, CINAHL, arXiv, CiteSeerX, IEEEXplore) using search terms 'nursing' (for computer science, maths and engineering only as this is a minority term), 'nurse staffing', 'nurse workload', 'hospital staffing', 'nursing ratios', 'safe staffing', 'staffing' and 'patient safety' in the past 15 years (2003–2018). All terms were in English. A successive fractions approach was used to refine this search.[10] Systematic reviews were included where primary research is available. Initial reading also revealed a paucity of work in area such

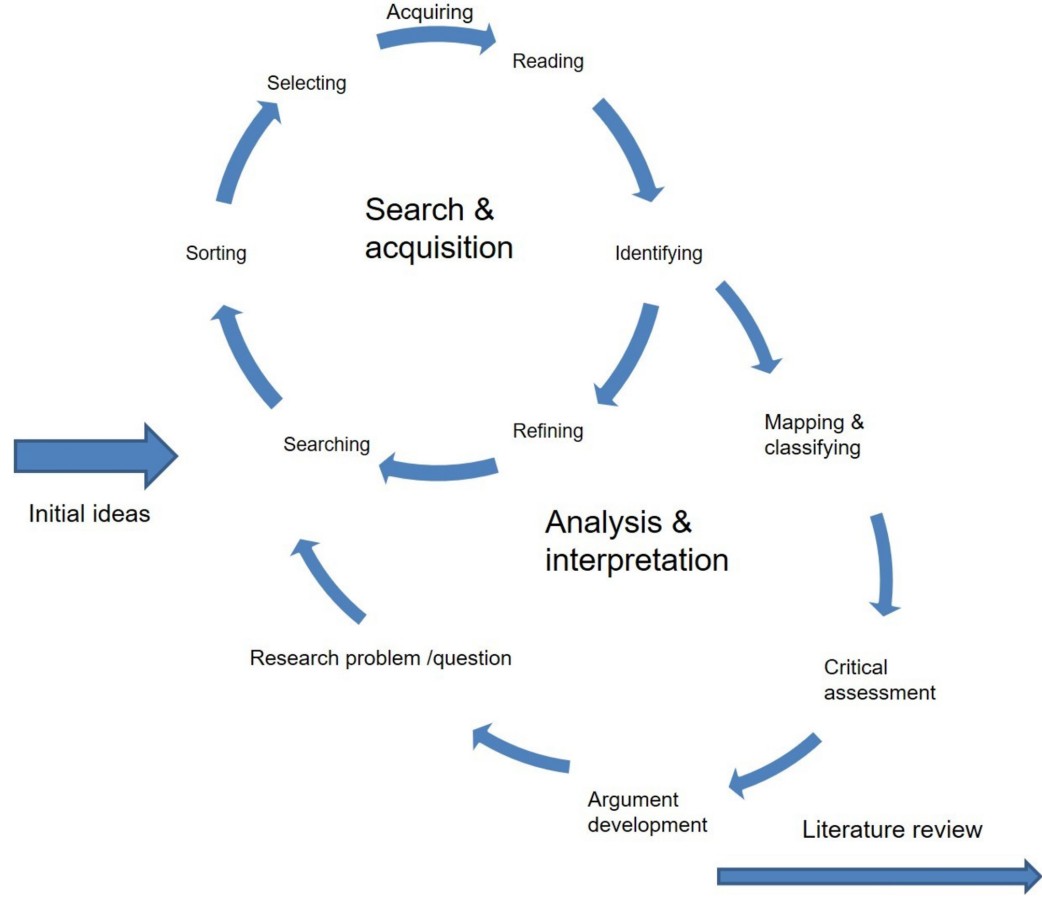

**Figure 1** The hermeneutic circle as a framework for literature reviews.

as mental health, community care and learning disability and so the focus remained on acute hospital staffing.

This generated 7323 items. A significant proportion of work in the area of nurse staffing has taken the form of editorials and various types of commentary or opinion pieces rather than empirical studies. After elimination of opinion pieces and professional non-peer-reviewed literature, there remained in excess of 700 papers (n=769) across the different disciplines. These were reviewed and have broadly been themed to gain insight into the different approaches and relational research used to examine the issue of staffing within a wider context of safety. An online supplementary file contains more detail on how these studies were selected.

### Patient and public involvement
This is a review of the existing literature and so there was no patient involvement at this stage. Subsequent iterations would have patients/public coresearchers to cocreate questions and design subsequent studies.

### RESULTS
In terms of analysis, specific themes started to emerge which were anchored around methodology and the different lenses though which the issue of nurse staffing is seen. Approaches to the question framed as 'how many do we need?' are commonly approached as units, for example, numbers of workforce, ratios (nurse to patient or nurse to furniture) unit/ward levels of activity, job (different types of nursing such as paediatric, theatres or intensive care) or contextually/situationally. It is interesting to note that some situational perspectives originated from systems engineering approaches such as Carayon and Gurses[12] systems engineering approach in intensive care units.

After examining the literature iteratively, a number of themes emerge. Any of these themes would merit study in their own right, but a broad overview of the different approaches and their application is given here. What is striking is the range of approaches used. These range from descriptive studies to the development of operational mathematical models.

### Nursing as units, workload or activity
The literature on the development of a ratio or the effect of nurse to patient ratios is a common theme in the nursing literature. A meta-analysis by Driscoll et al[13] draws on 35 studies which show an association between nurse staffing ratios and patient outcomes. These studies use large administrative datasets and found that higher staffing levels were associated with reduced mortality, reduced medication errors, reduction in incidence of pressure ulcers, reduced use of restraint, reduction in infections, such as hospital-acquired pneumonia, higher aspirin use and a greater number of patients receiving treatment. It is interesting to note that all of these studies are either cross-sectional or point prevalence studies.

When examining this section of the literature, cross-sectional studies dominate.

Shindul-Rothschild et al[14] note how workload impacts on efficiency in emergency departments. Using regression in trauma centres, the time to a diagnostic evaluation significantly increased when nurses cared for a higher number of patients. Aikens work on RN 4Cast[15] shows that an increase in a nurses' workload by one patient increased the likelihood of an inpatient dying within 30 days of admission by 7%. They also found that every 10% increase in bachelor's degree nurses was associated with a decrease in this likelihood by 7%. These associations imply that patients in hospitals in which 60% of nurses had bachelor's degrees and nurses cared for an average of six patients would have almost 30% lower mortality than patients in hospitals in which only 30% of nurses had bachelor's degrees and nurses cared for an average of eight patients. Skillmix is also a feature of this body of work. Graduate RNs are associated with better outcomes while higher RN to patient ratios or support worker to RN ratios are associated with poorer outcomes.[16] The study by Boyle et al[17] found that higher registered nurse hours per patient day was associated with improvements in total fall rates over time. This finding was not unique with various studies exposing the same association, for example, Staggs and Dunton[18] found that skillmix of a higher rate of assistive personnel was associated with an increase in falls but there was variation. On acute medical units, there was a weak association between decrease in falls and increase in RN staffing, but in step down and medical units the authors concluded that increasing non RN staffing was ineffective for decreasing falls. Increasing RN staffing did have a positive impact but this varied by unit type. White et al[19] in another cross-sectional study (n=353 333) examined a historical dataset using 30-day mortality and failure to rescue as endpoints and found that each 10% increase in the proportion of graduate nurses was associated with 4% lower odds of death in the older population 10% lower odds of death for those with dementia. This was associated with better odds of rescue where graduate nurses are deployed.

Relationships between staffing factors and outcomes are complex[20–22] and the causal relationship is not fully understood. However, we know from other studies that there is a deficit in care if staffing is not adequate. For example, in a study by Ball et al,[23] nurses (86%) reported that one or more care activity had been left undone due to lack of time on their last shift. Most care frequently left undone were comforting or talking with patients (66%), educating patients (52%) and developing/updating nursing care plans (47%). The odds of care being left undone halved when nurses had six or less patients to care for. The work left undone has become a more specific area of enquiry with more authors coming to essentially the same conclusions.[24] Missed care appears to be common and perhaps even predictable. The work of Bragadottir et al[25] examined the correlates and predictors of missed nursing care in hospitals using regression.

Such approaches may contribute to future demand-based models.

A substantial body of evidence has been built which uses the approach of nurses, nursing care or patients as units of work activity.

Many factors contribute to workload and the relationship with safety is frequently investigated. Adequate staffing and resources, administrative support and collaborative workplace relationships have been shown to improve patient safety, and factors such as low job satisfaction, staff turnover and high workload increase risk to patient safety.[26 27]

Nursing workload is subjected to 'measurement' using a plethora of measurement tools which numerate tasks. Most of these tools tend to view nursing workload as countable tasks, which is likely to be an oversimplification of complex work. Such tools do not recognise the 'other' work which nurses do related to workplace culture and climate.[28] Fasoli *et al*[29] after an extensive review of nurse workload classification systems reported there was no gold standard system for doing this and current measures were not sensitive enough. This is reflected in findings that the data collected routinely by nurses is not of sufficient quality to perform such complex modelling.[18 30 31]

There has been a preoccupation with 'time and motion' studies but these are of limited use in complex work[32 33] as a result this method cannot handle relational work[34] and is therefore likely to underestimate nursing workload.

There are measures which consider the complexity of the work[35] and workload emerged as a theme including systematic reviews of the effect of workload on patient safety.[12 36] Several inductive subgroups emerged including the role of workload and subsequent delegation of tasks which leads to delegation of safety critical activity such as vital signs monitoring, for example, how local modifications to track and trigger systems can reduce accuracy the of predictive algorithms[37] or the workplace environment as a factor of workload.[38]

There has been some examination of redistribution of workload[39] in which there has been some measurable increase in clinical time by workload redistribution. Although this review does not consider the costs of staffing, there are some interesting papers on this. Perhaps one of the most interesting is Newbold's 2008 model[40] which used the study by Aiken *et al*[41] looked at nurse levels of education and patient mortality in terms of a trade-off: what would a cost/mortality look like. Other researchers have also utilised the cross-sectional work to iterate mathematical models with some success.

## Workforce

Levels of workforce skill and education also feature in the literature. Much of the work linking staffing (units or education) to outcomes looks at specific aspects of care or harm. Boyle *et al*[17] found an association between specialty nursing education (postregistration certification) and improvement in the rates of falls using a longitudinal model (903 hospitals over 6 years) which echoed the findings of Kendall-Gallagher and Blegen,[42] Lange *et al*[43] and Boltz *et al*.[44] For example, the Lange paper found that units staffed with two or more geriatric-certified nurses had significantly lower fall rates than units with one or no geriatric-certified nurses.[43] There appears to be a body of evidence supporting a higher skilled educated workforce as associated with less harm, although it is interesting to note that Magnet hospitals were associated with higher rates of falls.[45] There are numerous papers linking RN staffing with patient outcomes/harm but fewer on education. One interesting single-centre study looked at the consequences of outlying patients to non-specialist wards which was associated with an increase in mortality.[46]

In a recent systematic review,[47] 18 studies were examined which gave subjective reports of missed care. Seventy-five per cent or more nurses reported omitting some care. Fourteen of these studies found low nurse staffing levels were significantly associated with higher reports of missed care. There was little evidence that adding support workers to the team reduced this. The authors noted that the extent to which the relationships observed link to outcomes has yet to be investigated.

Other work suggests that adding assistive nursing personnel without professional nurse qualifications may contribute to preventable deaths, erode quality and safety of hospital care.[15] Where studies have explored the impact of second-level nurses, similar to enrolled nurse qualification, the evidence is not supportive of the role.[48]

Satisfaction with work and other factors such as environmental, workload, working hours and effect of skill mix is also reported as a factor. In some cases, there is evidence of an association between shift length, staff satisfaction and patient outcomes[49] which associated longer shifts with more care left undone. These studies usually take the form of cross-sectional surveys as there appears to be no routine data collection of these data. Unlike the concepts of staffing, there is much more consistency in the reporting of issues related to workload such as moral distress, burn out and compromise of both personal and professional values including the erosion of identity.[50] The corollary of this also occurs in that papers describing moral distress also cite inadequate staffing as a causative attribute.[51]

Staffing and turnover are also occasionally reported in terms of outcomes, for example, turnover and patient outcomes. In 2014, Park *et al*[52] examined episodic unit-level data from 2008 to 2010. This study examined 10935 unit-quarter observations (2294 units, 465 hospitals) using multilevel regression. They found that the effect of RN turnover on unit-acquired pressure ulcers was significant and 'lagged' in terms of time. For every 10% point increase in RN turnover in a quarter, the odds of a patient having a pressure ulcer increased by 4% in the next quarter. Higher RN turnover in a quarter was associated with lower RN staffing in the current and subsequent quarters. Higher RN staffing was associated with lower pressure ulcer rates, but it did not mediate the relationship between turnover and pressure ulcers.

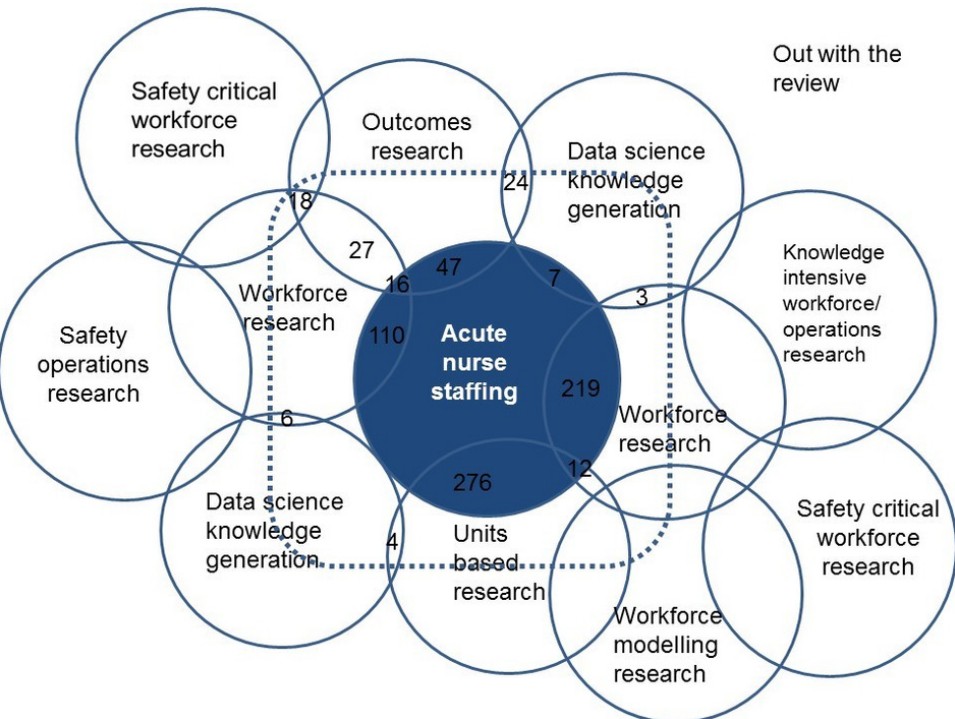

**Figure 2** A Venn diagram of areas of enquiry and potential overlap with other interdisciplinary areas—the bordered area contains the literature reviewed.

There appears to be a growing body of literature which examines resilience. The emotional labour of nursing involves managing the emotional demands of nursing work. An integrative review by Delgado *et al*[27] identified the difficult nature of not only the work but also the opportunities, or lack of, to build personal resilience.

Dos Santos Alves *et al*[53] found that RNs nurses with greater autonomy, good working relationships and control over their work environment had lower levels of emotional exhaustion, higher job satisfaction and less intention of leaving. Although they did not associate findings with patient outcomes, these and other authors have found positive associations with autonomy and rescue rates and mortality and they are more likely to experience this in small and non-teaching hospitals.[54] This a consistent theme in the literature reflected in a scoping review of 12 studies[55] which concludes that structural empowerment effects the quality of care and patient safety in hospital. This is also reflected in perceptions of safety culture and outcomes. When adequate resource is allocated in terms of staffing the perception of safety and patient satisfaction improves.[56]

### Understanding the complex relationship

There were a number of studies that used various aspects of data science. These ranged from Bayesian approaches,[57] to systems thinking, modelling, computational mathematics and approaches such as machine learning. Many of these papers appeared outside of the nursing literature and were located via databases that serve the physical sciences, engineering and mathematics.

This includes work such as Aickelin *et al*,[58] who developed a memetic evolutionary algorithm to achieve explicit learning in rule-based nurse rostering, which involves applying a set of heuristic rules for each nurse's assignment. This uses a set of building blocks in terms of data and rules to build an estimation of distribution algorithm. As the authors point out, although this performs well in some 'real-world' situations, it is limited by its ability to learn—one of the solutions is to add more nurses to the model without understanding that this might be a constraint.

Pitkäaho *et al*[59] used Finnish data from over 35 000 episodes of care to determine relationships between nurse staffing and patients' length of stay in acute care units and to determine whether non-linear relationships exist between variables using a Bayesian approach. They found that acuity was the overriding factor that connected all 11 variables in the dependency network of nurse staffing and short length of stay. Non-linear associations were found between short length of stay and the proportion of RNs. Skill mix consisting of an average proportion of RNs (65%–80%) was conducive to a short length of stay and predicted a 66% likelihood of short length of stay. Lower percentages of RN predicted lower likelihood of short length of stay.

An overriding theme in these approaches are challenges in the real world. Well-constructed approaches to calculating staffing needed more work than authors anticipated when tested in reality. This is not unusual—these approaches tend to be iterative; however, it is interesting to note that a number of these approaches concluded

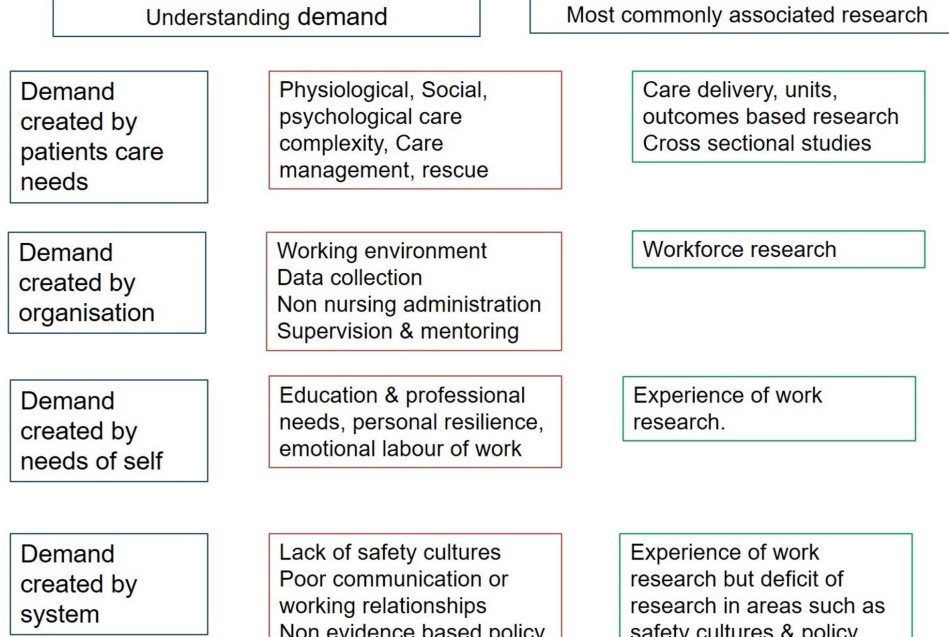

**Figure 3** Contextualising the findings in terms of demand.

that previous staffing models had underestimated ratios and staffing requirements in areas examined such as recovery.[60]

Other authors such as Park[61] are now building more optimised models based on operational mathematical approaches and are likely to yield a more comprehensive approach to the problem of computing staffing and outcomes as they accommodate complexity.

Data science approaches were the only ones to consider knowledge stock or knowledge flow in the early part of the 21st century but little consideration was given to this after this time.[62]

The advent of data science offers many opportunities; however, nursing may not be placed to capitalise on them.[63] Lack of high-quality data is a recurring issue in terms of both unit-based approaches and approaches using data science to examine complexity. Nursing data are generally episodic and lack sensitivity to the activity that nurses perform. In the studies, most informatics systems use taxonomies or lists of tasks which are limited and show no apparent relationship with time taken. Another issue is the dominance of supply side (nurse units of time, for example) and not the demand from patients which is rarely looked at.

### Refining and leaving the hermeneutic circle

A key facet of Gadamer's[9] approach is not to leave the hermeneutic circle due to its inescapability and as such the review has focused on the search. However, the authors are from a positivist paradigm and so has broken the circle at this juncture. This method is an inductive one and as such recognises that the iterations of searching and analysis could be infinite. The themes that emerge here across the literature are not exhaustive but provide a break in the cycle of searching and analysis. The exploration of these themes can be further built up. The papers reviewed offered many directions in which to expand this area of enquiry and these are shown in figure 2, a Venn diagram of the areas explored, the numbers of papers reviewed and potential for intersection with areas of knowledge not reviewed herein.

### DISCUSSION

Although the evidence in the nursing literature appears to offer no firm guidance on staffing models or absolute solutions, this could be seen as reassurance because it also demonstrates the complexity of the problem. The literature describes different associations between various factors such as outcomes and staffing numbers/skill mix. Each piece of work gives a slightly different perspective but an overarching emerging theme is that a relationship does exist between different factors even if these relationships are not fully understood, there is an apparent effect.

The conceptual difficulty safety presented in the context of staffing was an emerging theme. How is safe differentiated from unsafe, what is optimal staffing and where should trade-off occur. These studies help with clarification of the problem but there is little consistency in this body of work in terms of a solution.

One of the themes herein was the repeated association with not only numbers of RNs but also the educational level in the workplace. This appeared to show an overall benefit in employing RNs and also RNs with a degree level qualification. Some authors note this in the employment marketplace—that employers are acting on these findings and recording a downward trend in using assistive roles in the USA.[64]

It is interesting that only one paper mentioned knowledge stock and knowledge flow. None of the papers reviewed examined nurse staffing in terms of being a knowledge intense occupation which is a factor in modelling workforces in other safety critical industries or in other fields where operations are highly reliant on professional knowledge such information technology.[65] This might be because the approach to workforce modelling in nursing is focused on linear, deterministic approaches such as time and motion or time filled with tasks. This is more akin to workforce modelling in the service industries.[66]

There is narrative which focusses on a fixed ratio as staffing model in nursing. Ratios are common in other safety critical industries or area such as mass gatherings; however, it is often used as a failsafe rather than a staffing model[67] which might be a more practical option for nursing.

The literature is supportive of a relationship between staffing, skillmix and education and this has been reviewed before[3]; however, circling out beyond the nursing literature affirms this. Many reviews or policy documents appear to be confined to the nursing or medical literature and yet a rich seam of enquiry appears outside of these fields. What is striking is that there was very little overlap in areas of enquiry such as safety critical operations research, demand modelling or knowledge-based workforce research and acute nurse staffing (figure 2). These areas show in the Venn diagram (figure 2) as a representation of the sets of knowledge reviewed and those not reviewed specifically but where these different fields appear to intersect. Researchers should widen their perspectives on methodologies and approaches to include other disciplines particularly the approaches of safety critical industries. By doing so, it is possible to iterate an initial understanding of demand which can begin to integrate areas such as workforce and safety—an interpretation is given in figure 3.

### Strengths and limitations of this study

This is a review of the existing literature across many fields and focuses on cross-disciplinary approaches. By encompassing many fields, breadth rather than depth limits the analysis but does take the literature in context to form an overall view. There is a risk of over simplifying the literature and the knowledge at this scale.

### CONCLUSION

There is a wide variety of literature from different paradigms that support a complex interrelationship between different factors in acute nurse staffing. Despite a growing body of knowledge, there appears to be little reference to nursing as safety critical nor is the problem viewed through this lens. It is suggested that looking outside the discipline of nursing might add valuable insight to this problem.

The issue of nurse staffing is a complex one and the relationships between factors such as nursing and patient outcomes is also complex. Despite this, no papers examined nursing as a knowledge intensive operation or as a safety critical workforce. There is an increasing body of knowledge outside of nursing which has focused on this topic but is rarely utilised.

Although this is an emerging area, evidence repeatedly suggested a complex interdependent relationship between nurse staffing and various factors such as patient safety. Hermeneutic approaches can offer new insight by focussing in interpretation and has been used to generate knowldege.[68]

Given the asserted complexity of work, time and motion or other simplistic activity analysis (measuring nursing in a linear or deterministic way) should be avoided. The importance of this emerging relationship should be considered when examining staffing. This study suggests looking at the issue of safe staffing from a wider operational safety perspective could uncover additional insight and solutions.

**Twitter** @alisonleary1

**Acknowledgements** The authors would like to acknowledge Dr M Alexander for help in identifying cases from the coroner. Dr Elaine Maxwell for help in formulating the area of enquiry.

**Contributors** AL design, analysis and writing. GP analysis and writing.

**Funding** The authors have not declared a specific grant for this research from any funding agency in the public, commercial or not-for-profit sectors.

**Competing interests** None declared.

**Patient consent for publication** Not required.

**Ethics approval** This is a review of the literature. Ethical permission was not sought.

**Provenance and peer review** Not commissioned; externally peer reviewed.

**Data sharing statement** All of the papers reviewed are in the public domain.

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
