## [Reviewer comments · BMJ Open]

ARTICLE DETAILS

TITLE (PROVISIONAL)	Determining acute nurse staffing, a hermeneutic review of an evolving science.
AUTHORS	Leary, Alison; Punshon, Geoffrey

VERSION 1 - REVIEW

REVIEWER	Alejandra Recio Saucedo Research Fellow, NIHR Evaluation, Trials and Studies Coordinating Centre (NETSCC)
REVIEW RETURNED	19-Aug-2018

GENERAL COMMENTS	Review of the manuscript: Determining acute nurse staffing, a hermeneutic review of an evolving science This is a hermeneutic review (a theory dealing with close academic engagement and critical interpretive synthesis of key texts) of the research on methods to calculate nurse staffing and skill mix from multiple disciplines. The review includes the latest evidence on issues related to nurse staffing ratios, levels, education and solutions informed by theoretical, mathematical, computational or organisational models. The manuscript does an excellent job reflecting upon a large body of evidence. It is an important contribution to the field because the shortage of nursing staff is not likely to ease in the short- or medium- term and as such, it is imperative to collate, organise and develop the evidence available on the issues related to the delivery of health care linked to nurse staffing and skill mix. Searching for evidence outside health, such as informatics and computer science is forward-thinking as the solutions to the complex problems of nurse staffing, skill mix, patient safety, resource distribution, quality of care and patient outcomes, necessarily require an interdisciplinary approach. Solutions built upon engineering, mathematical and health care models are more likely to address the issues with strategies that are ethical, feasible and cost-effective. As the authors suggest, one field alone cannot solve the problem due to its inherent complexity and the need of specialist knowledge to generate, collect and interpret data/evidence. I understand that the authors' purpose was to explore the breadth of evidence making the inclusion of a study in emergency settings appropriate for the review (p5-l53). However, it is important to recognise that acute care and emergency care are too distinct to be compared side-to-side in terms of staffing issues and their
--

relationship to patient, organisation or staff outcomes. In relation to exploring nurse staffing in contexts other than acute care, the objectives of the review indicated the lack of attention that other contexts have received (e.g. nursing in community and mental health) but this topic was not further explored in the manuscript. Perhaps not enough evidence was found? In which case, the paucity of studies looking at staffing in settings outside acute hospital is in itself a relevant finding. Alternatively, if the objective of the review was to explore only nurse staffing in acute care, as per title, I suggest rewriting the objectives to clarify the issue explored.

The authors recognise that focusing on breadth rather than depth limited their analysis. I believe that in some instances, this approach resulted in oversimplification of some areas (see for instance p7-l23; p9-l28; p10-l12). Elaborating further on the studies utilising evolutionary algorithms, which failed to understand that limited resources were a constraint to a solution proposed, or the concepts of knowledge stock and flow in studies exploring nursing as a knowledge intense occupation, would be an important and original contribution to the literature.

For replicability of the study, it is advisable to elaborate on the mechanisms to filter down the evidence from 7323 to 767 studies. In addition, details on the tools and procedures used in conducting the mapping, classification and critical assessment of the evidence are not provided. How were disagreements in interpretation of the evidence resolved? How did the mapping and classification evolved throughout analysis?

Figure 3. I found the diagram a good effort to visually represent the categories/structure of the evidence reviewed, however, in its current form, it does not deliver a complete message. For instance, assuming that no-number in the intersection areas represent 1 study, would I add 1 from the upper-left corner of the diagram to 1 study from the lower-right corner for the intersections of Safety critical workforce research and Workforce research? I believe this is not the case, but the diagram draws me to the intersection areas that are blank v the ones with figures and to the duplicate circles, when the focus should be on the variability of studies available per area or intersecting areas.

Comments of style

There are various places in the text where sentences seem to stop mid-way, commas would enhance the flow, and where direct references in lieu of demonstrative pronouns would be helpful to the reader (p7-l29: "... can reduce accuracy the of predictive algorithms"; p6-l29: "...with 4% lower odds of death in the older population 10% lower odds of death for those with dementia"; p9-l28: "This includes works such as Aickeling ... ")

The results are written in a style that moves between present and past in ways that helped me to differentiate between the evidence from the studies and the analysis from the authors. However, this style is not consistent throughout.

The conclusion focused on one of the key points for practice that emerged from the review, however, the point made about looking at the body of knowledge outside of nursing was not elaborated

	upon. Perhaps the author did not consider this to be of sufficient relevance? I believe that a sub-header got printed as opening sentence of paragraph (p8-l25: "Satisfaction with work and other factors ...") The section Refining and leaving the hermeneutic circle might be better positioned earlier in the manuscript.
--	--

REVIEWER	Liz Smythe Auckland University of Technology (AUT) New Zealand
REVIEW RETURNED	13-Sep-2018

GENERAL COMMENTS	I really enjoyed reading this paper and believe it is a valuable contribution to the literature. I believe it is fine as it stands. My comments are as a hermeneutic researcher for your ongoing considerations. It would be fascinating to do a review of all the opinion pieces. Your current review is about what can (or cannot) be measured. That immediately limits the possible meanings. Opinion pieces may well capture the more complex, nuanced wisdom that comes from lived experience. In my experience of once being a nurse, you could walk onto any ward and immediately sense the workload. It did not need any sort of analysis. A too-busy ward 'screamed' as bells went unanswered, nurses scurried with worried looks on their faces, no one was 'around' to talk to. Have there been qualitative studies that capture this sort of knowing? The voice that seems to be missing is the patients. What is their experience of there not being enough staff? There was one mention of 'patient satisfaction' but again this was likely measured. I was interested that the nursing assistants did not rate well. I have had several personal experiences of late when it has been the nursing assistant who attending to hygiene needs as brought much needed 'care' to the patient. It seemed they were often the only ones known by name, the only ones with whom the patient could build a relationship. Do such things get covered in the measures? Here are a couple of my favourite Gadamer quotes: Gadamer, H. G. (1982). Truth and method (G. Barden & J. Cumming, Trans.). New York: Crossroad. "The truly experienced man is one who is aware of this, who knows that he is master neither of time nor the future. The experienced man knows the limitedness of all prediction and the uncertainty of all plans. In him is realised the truth value of experience" (p.320) "Insight is more than knowledge of this or that situation. It always involves an escape from something that had deceived us and held us captive" (p.319-320) I encourage you to get more hermeneutic in your ongoing explorations. It is the missing voice. I believe it is that hermeneutic voice that needs to guide us forward.
--

VERSION 1 – AUTHOR RESPONSE

Reviewer One

Thank you for your helpful feedback-we have amended the MS.

This is a hermeneutic review (a theory dealing with close academic engagement and critical interpretive synthesis of key texts) of the research on methods to calculate nurse staffing and skill mix from multiple disciplines. The review includes the latest evidence on issues related to nurse staffing ratios, levels, education and solutions informed by theoretical, mathematical, computational or organisational models.

The manuscript does an excellent job reflecting upon a large body of evidence. It is an important contribution to the field because the shortage of nursing staff is not likely to ease in the short- or medium- term and as such, it is imperative to collate, organise and develop the evidence available on the issues related to the delivery of health care linked to nurse staffing and skill mix.

Searching for evidence outside health, such as informatics and computer science is forward-thinking as the solutions to the complex problems of nurse staffing, skill mix, patient safety, resource distribution, quality of care and patient outcomes, necessarily require an interdisciplinary approach. Solutions built upon engineering, mathematical and health care models are more likely to address the issues with strategies that are ethical, feasible and cost-effective. As the authors suggest, one field alone cannot solve the problem due to its inherent complexity and the need of specialist knowledge to generate, collect and interpret data/evidence.

I understand that the authors' purpose was to explore the breadth of evidence

making the inclusion of a study in emergency settings appropriate for the review (p5-l53). However, it is important to recognise that acute care and emergency care are too distinct to be compared side-to-side in terms of staffing issues and their relationship to patient, organisation or staff outcomes.

Agree-others have also argued in the acute setting that other specialisms are unique for example older peoples care. As we were looking at breadth and these papers met the criteria we decided to review them. We do not seek to make comparisons so kept the work in.

In relation to exploring nurse staffing in contexts other than acute care, the objectives of the review indicated the lack of attention that other contexts have received (e.g. nursing in community and mental health) but this topic was not further explored in the manuscript. Perhaps not enough evidence was found? In which case, the paucity of studies looking at staffing in settings outside acute hospital is in itself a relevant finding. Alternatively, if the objective of the review was to explore only nurse staffing in acute care, as per title, I suggest rewriting the objectives to clarify the issue explored.

The authors recognise that focusing on breadth rather than depth limited their

analysis. I believe that in some instances, this approach resulted in

oversimplification of some areas (see for instance p7-l23; p9-l28; p10-l12).

Elaborating further on the studies utilising evolutionary algorithms, which failed to understand that limited resources were a constraint to a solution proposed, or the concepts of knowledge stock and flow in studies exploring nursing as a knowledge intense occupation, would be an important and original contribution to the literature.

Very much agree and propose this as a subsequent piece of work thus have added it to the discussion/conclusion.

For replicability of the study, it is advisable to elaborate on the mechanisms to filter down the evidence from 7323 to 767 studies. In addition, details on the tools and procedures used in conducting the mapping, classification and critical assessment of the evidence are not provided. How were disagreements in interpretation of the evidence resolved? How did the mapping and classification evolved throughout analysis?

Thank you we have now included a supplementary file. We used the Boell & Cecez-Kecmanovic paper primarily looking at units, disciplines etc. As first time authors in the filed the conventions of writing this kind of review are not entirely known to us.

Figure 3. I found the diagram a good effort to visually represent the categories/structure of the evidence reviewed, however, in its current form, it does not deliver a complete message. For instance, assuming that no-number in the intersection areas represent 1 study, would I add 1 from the upper-left

corner of the diagram to 1 study from the lower-right corner for the intersections of Safety critical workforce research and Workforce research? I

believe this is not the case, but the diagram draws me to the intersection areas that are blank v the ones with figures and to the duplicate circles, when the focuss should be on the variability of studies available per area or intersecting areas.

This is most defiantly a first iteration and perhaps a reflection of our own backgrounds in math (demand modelling) and very much a response to policy makers to this body of evidence, which although growing is rarely reviewed in the policy context (hence the contextual aspect). Figure 3 is something that very much needs further development. What we have tried to do is capture the completeness of the knowledge in the texts (difficult for a positivist used to a more reductionist approach!) so may well seem quite clumsy. Apologies for this.

Comments of style

There are various places in the text where sentences seem to stop mid-way,

commas would enhance the flow, and where direct references in lieu of

demonstrative pronouns would be helpful to the reader (p7-l29: "... can reduce accuracy the of predictive algorithms"; p6-l29: "...with 4% lower odds of death in the older population 10% lower odds of death for those with dementia"; p9-l28: "This includes works such as Aickeling ... ")

The results are written in a style that moves between present and past in ways that helped me to differentiate between the evidence from the studies and the analysis from the authors. However, this style is not consistent throughout.

Thank you this has been addressed

The conclusion focused on one of the key points for practice that emerged from the review, however, the point made about looking at the body of knowledge outside of nursing was not elaborated upon. Perhaps the author did not consider this to be of sufficient relevance?

Agree It is very relevant-we have given it more emphasis.

I believe that a sub-header got printed as opening sentence of paragraph (p8-l25:

“Satisfaction with work and other factors ...”)

The section Refining and leaving the hermeneutic circle might be better positioned earlier in the manuscript.

Reviewer Two

Many thanks for the considerate and helpful feedback-it really was a delight to read also the Gadamer quotes. We are very much positivists and this is a new approach for us and so such support is very welcome.

I really enjoyed reading this paper and believe it is a valuable contribution to the literature.

I believe it is fine as it stands.

My comments are as a hermeneutic researcher for your ongoing considerations.

It would be fascinating to do a review of all the opinion pieces. Your current review is about what can (or cannot) be measured. That immediately limits the possible meanings. Opinion pieces may well capture the more complex, nuanced wisdom that comes from lived experience. In my experience of once being a nurse, you could walk onto any ward and immediately sense the workload. It did not need any sort of analysis. A too-busy ward ‘screamed’ as bells went unanswered, nurses scurried with worried looks on their faces, no one was ‘around’ to talk to. Have there been qualitative studies that capture this sort of knowing?

Agree this would be very useful its beyond the scope of work here but would be fascinating (we would need to collaborate on such as study as we are data scientists/mathematicians!)

The voice that seems to be missing is the patients. What is their experience of there not being enough staff? There was one mention of ‘patient satisfaction’ but again this was likely measured.

We expected to find more on this but there was very little.

I was interested that the nursing assistants did not rate well. I have had several personal experiences of late when it has been the nursing assistant who attending to hygiene needs as brought much needed ‘care’ to the patient. It seemed they were often the only ones known by name, the only ones with whom the patient could build a relationship. Do such things get covered in the measures?

Generally outcomes are predetermined by the researchers (and perhaps funders) ie survival and so do not emerge it would be interesting to look at others perceptions-one thing often debated is what is “safe” we know a lot about safety as the absence of harm but almost no attention is paid ie to comfort or psychological safety for example.

Here are a couple of my favourite Gadamer quotes:

Gadamer, H. G. (1982). *Truth and method* (G. Barden & J. Cumming, Trans.). New York: Crossroad.

"The truly experienced man is one who is aware of this, who knows that he is master neither of time nor the future. The experienced man knows the limitedness of all prediction and the uncertainty of all plans. In him is realised the truth value of experience" (p.320)

"Insight is more than knowledge of this or that situation. It always involves an escape from something that had deceived us and held us captive" (p.319-320)

I encourage you to get more hermeneutic in your ongoing explorations. It is the missing voice. I believe it is that hermeneutic voice that needs to guide us forward.

Thank you-its been a really interesting (if very labour intensive!) experience. We admire your persistence!

VERSION 2 – REVIEW

REVIEWER	Alejandra Recio Saucedo NIHR Evaluation, Trials and Studies Coordinating Centre (NETSCC), University of Southampton United Kingdom
REVIEW RETURNED	09-Dec-2018

GENERAL COMMENTS	Dear Authors Thank you for the resubmission of your manuscript. It is truly an enjoyable read and a good contribution to the literature. You have addressed most of the points in the first submission and the notes I am adding here are focused on increasing readability of your manuscript and emphasising the contribution of your study to the methodology of conducting review studies. Manuscript A few more modifications to Figure 2 would enhance the visual. Visualisation theories and guides suggest that every element in figures or graphs be meaningful, and if an element (colour, size, type of graph) is not providing meaning, then a simpler diagram would be better suited to represent abstract concepts (Storytelling with data by Cole Nussbaumer Knaflic or Edward Tufte’s data-ink ratio concept are great resources). In Figure 2, the size of the intersection areas is not consistent with the number of studies that were found to overlap between acute nurse staffing and other fields. Circles of different sizes might be appropriate to show the ‘size’ of the overlap. In addition, the colour scheme does not seem to add meaning to the argument presented by the authors, which encourage researchers to widen perspectives in their enquiries. Were there overlapping areas that require more attention than others? Or some that were more promising? There is information in the text about this that does not translate to the figure. A way to answer these questions could be shading only some of the circles and leaving the less meaningful ones in black and white. Finally, what is the meaning of the intersection in the circles outside the bordered area? (e.g. Safety critical operations research and Safety critical workforce research)? Supplemental material The supplemental material that provides details of how the evidence collected was processed is useful. Thank you for preparing this. A question about the number of sources reported: are totals of the first iteration 7,386 studies and not 7,323? Were there 2,058 studies included? 2,054 studies of which 1,285 were excluded, correctly shows the 769 sources analysed in the review. Also, how were disagreements in interpretation of the evidence to include/exclude
---

	resolved? How did the Boell & Cecez-Kecmanovic paper looking at units, disciplines inform your analysis? It would be very interesting to read for instance how the analysis of the studies found in the first search was used to inform new searches: which concepts, terms or phrases were useful in identifying relevant literature? Which concepts did not yield useful results? A narrative describing that process would be a strong contribution from your study, especially helpful if someone would like to repeat your method to explore the overlapping areas identified. The following sentence in the material: “With the further application for example of Łoś’s Theorem, and ultraproduct could eventually be defined.” is not fully formed (or perhaps this results of a typo in “and” which may have been intended to be “an”). Since this is supplemental material with room to provide further information on theories used in the study, it would be possible to expand the concept of the theorem and its mechanisms for “gap spotting”. Novel methodological approaches are strong contributions of review studies and at present, there is still little information on the way that 769 studies were analysed and categorised into areas. Style and references p8-125: “Satisfaction with work and other factors ...” The opening sentence of the paragraph reads like a sub-header. P9-130 “Baysian” P10-130 “Another issue is the dominance of supply side (nurse units of time for example) and not the demand from patients which is rarely looked at.” It would be worth mentioning studies that have explored the use of tools like Rafaela or the Safer Nursing Care Tool. P10-139 “However the author ...” The study was written by two authors.
--	--

REVIEWER	Elizabeth Smythe Auckland University of Technology New Zealand
REVIEW RETURNED	22-Nov-2018

GENERAL COMMENTS	I think this is an excellent paper. I commend you in tackling this issue in such a discerning manner.
---

VERSION 2 – AUTHOR RESPONSE

Reviewer(s)' Comments to Author:

Reviewer: 2

Reviewer Name: Elizabeth Smythe

Institution and Country: Auckland University of Technology, New Zealand

Please state any competing interests or state 'None declared': None declared

Please leave your comments for the authors below

I think this is an excellent paper. I commend you in tackling this issue in such a discerning manner.

Thank you

Reviewer: 1

Reviewer Name: Alejandra Recio Saucedo

Institution and Country: NIHR Evaluation, Trials and Studies Coordinating Centre (NETSCC),
University of Southampton, United Kingdom

Please state any competing interests or state 'None declared': None declared

Please leave your comments for the authors below

Dear Authors

Thank you for the resubmission of your manuscript. It is truly an enjoyable read and a good contribution to the literature. You have addressed most of the points in the first submission and the notes I am adding here are focused on increasing readability of your manuscript and emphasising the contribution of your study to the methodology of conducting review studies.

Thanks for taking the time to re-read this-it's obviously disappointing to us that you reverted from minor corrections to major revisions in a time sensitive paper. We have sought to address your comments below and your attention to detail is appreciated.

Manuscript

A few more modifications to Figure 2 would enhance the visual. Visualisation theories and guides suggest that every element in figures or graphs be meaningful, and if an element (colour, size, type of graph) is not providing meaning, then a simpler diagram would be better suited to represent abstract concepts (Storytelling with data by Cole Nussbaumer Knaflic or Edward Tufte's data-ink ratio concept are great resources). In Figure 2, the size of the intersection areas is not consistent with the number of studies that were found to overlap between acute nurse staffing and other fields. Circles of different sizes might be appropriate to show the 'size' of the overlap.

Thanks for your comments on this. As data scientists we understand the importance of visualisation. We have not made the nature of Fig 2 clear. This is simply a Venn diagram. We have therefore made it clearer to the reader that this is a Venn diagram and the numbers of papers in common elements in the intersection. Although it's difficult to determine the dimension of the set of elements it might be possible for others to then use this as the basis of further modelling.

In addition, the colour scheme does not seem to add meaning to the argument presented by the authors, which encourage researchers to widen perspectives in their enquiries.

We have taken out the colours. They were just to distinguish the common sets but otherwise had no significant meaning.

Were there overlapping areas that require more attention than others? Or some that were more promising?

No its just a Venn diagram used to describe what we found in the literature and where others might go next. We have made this clearer in the text. No other meaning should/can be inferred from this

diagram or this work. It's an area for further research. Edwards "Cogwheels of the Mind" is a great book on Venn diagrams (pub 2004).

There is information in the text about this that does not translate to the figure.

Thanks. We have now described in the text that it's a Venn diagram.

A way to answer these questions could be shading only some of the circles and leaving the less meaningful ones in black and white. Finally, what is the meaning of the intersection in the circles outside the bordered area? (e.g. Safety critical operations research and Safety critical workforce research)?

We have taken out all colour and shading and left this simply as a Venn diagram.

Supplemental material

The supplemental material that provides details of how the evidence collected was processed is useful. Thank you for preparing this.

A question about the number of sources reported: are totals of the first iteration 7,386 studies and not 7,323? Were there 2,058 studies included?

2,054 studies of which 1,285 were excluded, correctly shows the 769 sources analysed in the review.

769 were included as stated in the text. We have made how we go to this number clearer in the sup material. 2054 was a typo thanks!

Also, how were disagreements in interpretation of the evidence to include/exclude resolved? How did the Boell & Cecez-Kecmanovic paper looking at units, disciplines inform your analysis?

As this is described in the method, it seems redundant to explain it twice?

It would be very interesting to read for instance how the analysis of the studies found in the first search was used to inform new searches: which concepts, terms or phrases were useful in identifying relevant literature? Which concepts did not yield useful results? A narrative describing that process would be a strong contribution from your study, especially helpful if someone would like to repeat your method to explore the overlapping areas identified.

We have included these key items in the supplementary material and a narrative of how it was done is in the fairly detailed methods section and of course the results. It does not seem conventional to these reviews to do this and has not been mentioned previously in the review process? The emerging findings ie those in Figure 2 are now more linked to the text. In any paper it would be good to include more detail but the intention of the work is to look at the big picture.

The following sentence in the material: "With the further application for example of Łoś's Theorem, and ultraproduct could eventually be defined." is not fully formed (or perhaps this results of a typo in "and" which may have been intended to be "an").

Thank you "an" was the intended word this has been changed

Since this is supplemental material with room to provide further information on theories used in the study, it would be possible to expand the concept of the theorem and its mechanisms for "gap spotting". Novel methodological approaches are strong contributions of review studies and at present, there is still little information on the way that 769 studies were analysed and categorised into areas.

This did not appear to be a new methodological approach? We found many other hermeneutic reviews. Or are you referring to Los theorem? This is just a proposed approach to further modelling.

To generate an ultraproduct would require much more than the data available currently affords and might probably be a life's work! ie <http://mathworld.wolfram.com/LosTheorem.html>

We are very clear about where we break the circle

Style and references

p8-l25: "Satisfaction with work and other factors ..." The opening sentence of the paragraph reads like a sub-header.

Thank you. We have made this into a sentence.

P9-l30 "Baysian"

Thank you, we have corrected.

P10-l30 "Another issue is the dominance of supply side (nurse units of time for example) and not the demand from patients which is rarely looked at."

It would be worth mentioning studies that have explored the use of tools like Rafaela or the Safer Nursing Care Tool.

As with any paper much more detail could be provided including a description or critical analysis of these tools. Also these tools are not demand defining tools (though RAFAELA is close). These tools have been examined in detail elsewhere and it would be impossible to go into detail on over 700 papers. Why would the examination of two tools in particular add?

This study is aimed at a leadership as well as academic readership and so we have focussed on what we consider the over riding issue which is rarely examined-the dominance of supply side modelling.

P10-l39 "However the author ..." The study was written by two authors.

Thank you this has been changed.